# GradNorm: Gradient Normalization for Adaptive Loss Balancing in Deep Multitask Networks

## Abstract

Deep multitask networks, in which one neural network produces multiple predictive outputs, are more scalable and often better regularized than their single-task counterparts. Such advantages can potentially lead to gains in both speed and performance, but multitask networks are also difficult to train without finding the right balance between tasks. We present a novel gradient normalization (Grad-Norm) technique which automatically balances the multitask loss function by directly tuning the gradients to equalize task training rates. We show that for various network architectures, for both regression and classification tasks, and on both synthetic and real datasets, GradNorm improves accuracy and reduces overfitting over single networks, static baselines, and other adaptive multitask loss balancing techniques. GradNorm also matches or surpasses the performance of exhaustive grid search methods, despite only involving a single asymmetry hyperparameter $\alpha$. Thus, what was once a tedious search process which incurred exponentially more compute for each task added can now be accomplished within a few training runs, irrespective of the number of tasks. Ultimately, we hope to demonstrate that gradient manipulation affords us great control over the training dynamics of multitask networks and may be one of the keys to unlocking the potential of multitask learning.

## 1 Introduction

Single-task learning in computer vision has enjoyed much success in deep learning, with many models now performing at or beyond human accuracies for a wide array of tasks. However, a system that strives for full scene understanding cannot focus on one problem, but needs to perform many diverse perceptual tasks simultaneously. Such systems must also be efficient, especially within the restrictions of limited compute environments in embedded systems such as smartphones, wearable devices, and robots/drones. Multitask learning most naturally lends itself to this problem by sharing weights amongst different tasks within the same model and producing multiple predictions in one forward pass. Such networks are not only scalable, but the shared features within these networks tend to be better regularized and boost performance as a result. In the ideal limit, we can thus have the best of both worlds with multitask networks: both more efficiency and higher performance.

The key difficulty in multitask learning lies in the balancing of tasks, and perhaps the simplest way to control this balance is to choose the correct joint loss function. In practice, the multitask loss function is often assumed to be linear in the single task losses, $L = \sum_i w_i L_i$, where the sum runs over $T$ tasks. The challenge is then to find the best value for each $w_i$ that balances the contribution of each task for optimal model training. Our proposed method is furthermore an adaptive method, allowing $w_i$ to vary with the training step $t$, and so $w_i = w_i(t)$.

Our key insight lies in the observation that these $w_i(t)$ influence training only because they control the magnitude of the gradients generated from task $i$. As such, manipulating the gradient norms themselves would be a more direct way to control the training dynamics. More specifically, we propose a simple heuristic that penalizes the network when backpropagated gradients from any task are too large or too small. The correct balance is struck when tasks are training at similar rates; if task $i$ is training relatively quickly, then its weight $w_i(t)$ should decrease relative to other task weights

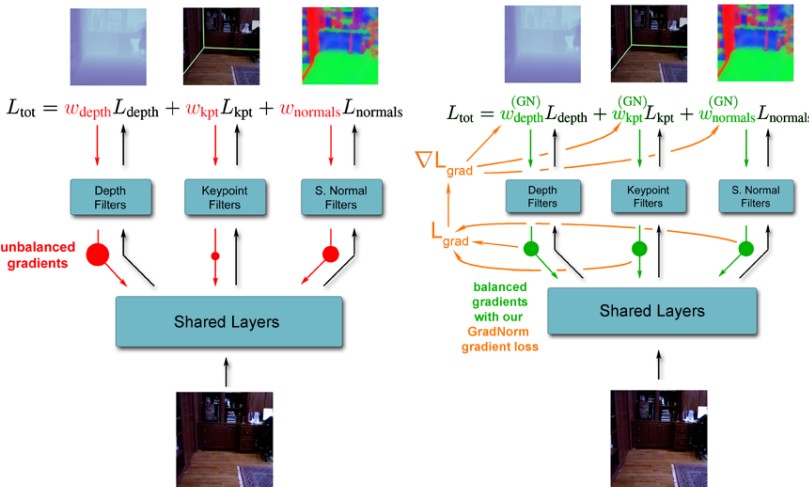

Figure 1: **Gradient Normalization.** Imbalanced gradient norms (left) result in suboptimal training within a multitask network, so we implement a novel gradient loss $L_{\text{grad}}$ (right) which detects such imbalances in gradient norms amongst tasks and tunes the weights in the loss function to compensate. We illustrate here a simplified case where such balancing results in equalized gradient norms, but in general some tasks may need higher or lower gradient norms relative to other tasks for optimal task balancing (discussed further in Section 3).

$w_j(t)|_{j \neq i}$ to allow other tasks more influence on the network. Our method can be said to be a form of batch normalization (Ioffe & Szegedy (2015)) for backpropagation, ensuring that gradients from each task per batch lie on a common statistical scale. We will show that, when implemented, gradient normalization leads to across-the-board improvements in accuracy and suppresses overfitting.

Our main contributions to the field of multitask learning are as follows:

1. An attractively simple heuristic for multitask loss balancing involving training rate equalization, which is implemented through a novel gradient loss function.

2. A simplification to exhaustive grid search (which has compute complexity $\mathcal{O}(N^T)$ for $N$ grid points in one dimension) that only involves tuning one robust hyperparameter.

3. Demonstration that direct interaction with gradients provides a powerful way of reasoning about multitask learning.

## 2 RELATED WORK

Multitask learning has existed well before the advent of deep learning (Caruana (1998); Bakker & Heskes (2003)), but the robust learned features within deep networks have spurned renewed interest. Although our primary application area is computer vision, multitask learning has applications in multiple other fields, from natural language processing (Hashimoto et al. (2016); Collobert & Weston (2008); Søgaard & Goldberg (2016)) to speech synthesis (Wu et al. (2015); Seltzer & Droppo (2013)), from very domain-specific applications like traffic prediction (Huang et al. (2014)) to very general cross-domain work (Bilen & Vedaldi (2017)).

Multitask learning is very well-suited to the field of computer vision, where making multiple robust predictions is crucial for complete scene understanding. Deep networks have been used to solve various subsets of multiple vision tasks, from 3-task networks (Eigen & Fergus (2015); Teichmann et al. (2016)) to much larger subsets as in UberNet (Kokkinos (2016)). Often, single computer vision problems can even be framed as multitask problems, such as in Mask R-CNN for instance segmentation (He et al. (2017)) or YOLO-9000 for object detection (Redmon & Farhadi (2016)). Researchers often assume a fixed loss function or network architecture, but there has also been significant work on finding optimal ways to relate tasks to each other in a multitask model. Clustering methods have

shown success beyond deep models (Kang et al. (2011); Jacob et al. (2009)), while constructs such as deep relationship networks (Long & Wang (2015)) and cross-stich networks (Misra et al. (2016)) search for meaningful relationships between tasks and learn which features to share between them. Work in Warde-Farley et al. (2014) and Lu et al. (2016) use groupings amongst labels to search through possible architectures for learning. Perhaps the most relevant to the current work, Kendall et al. (2017) uses a joint likelihood formulation to derive task weights based on the intrinsic uncertainty in each task.

## 3 METHODOLOGY

### 3.1 A GRADIENT LOSS FUNCTION BASED ON RATE BALANCING

We begin with the standard multitask loss function with time dependency, $L(t) = \sum w_i(t) L_i(t)$, and our goal is to learn the functions $w_i(t)$. We argued in Section 1 that $w_i(t)$ is intimately related to the norm of gradients from each task backpropagated into the network. We thus must motivate a set of desirable gradient magnitudes, and use those desired magnitudes to set the task weights $w_i(t)$.

Consider the norms of gradients from task $i$ on some set of weights $W$ within the network, $\text{norm}(\nabla_W L_i(t))$ (specific choices for $W$ to be discussed later). Our method of gradient normalization (hereafter referred to as GradNorm) works in two steps: (1) We first scale all gradient norms to an equal value as a neutral starting point. This value is most naturally chosen to be the average gradient norm amongst tasks, $E_{\text{task}}[\text{norm}(\nabla_W L_i(t))]$, where we use $E_{\text{task}}[X]$ to denote the average value of a task-dependent quantity $X$ across tasks. (2) We then modify gradient norms with a *rate balancing* term that ensures no task trains relatively too slowly. The gradient norms of task $i$ should grow when task $i$ trains relatively slowly, thereby boosting more sluggish tasks. Gradient norms thus should be an increasing function of the relative *inverse* training rate for each task.

To quantify training rates, we choose the *loss ratio* of task $i$ at training step $t$, $L_i'(t) := L_i(t)/L_i(0)$, as a measure of task $i$'s inverse training rate; smaller values of $L_i'(t)$ would mean that task $i$ has trained more. If $L_i'(t)$ denotes the inverse training rate of task $i$, then the *relative* inverse training rate is just $L_i'(t)/E_{\text{task}}[L_i'(t)]$. Using this simple loss ratio metric is valid for both regression squared loss and classification cross-entropy loss, as we will see in Section 5.2[1].

Our desired gradient norms are therefore:

$$\text{norm}(\nabla_W L_i(t)) \mapsto \text{(average gradient norm)} \times \text{(relative inverse training rate of task } i)^{\alpha}$$

$$= E_{\text{task}}[\text{norm}(\nabla_W L(t))] \left( \frac{L_i'(t)}{E_{\text{task}}[L_i'(t)]} \right)^{\alpha} \tag{1}$$

where $\alpha$ is an additional hyperparameter. $\alpha$ sets the strength of rate balancing in the multitask problem, and also is a measure of the *asymmetry* between tasks. In cases where tasks are very different in their complexity, leading to different learning dynamics, a higher value of $\alpha$ should be used to pull tasks back towards a common training rate more forcefully. When tasks are more symmetric (e.g. the synthetic examples in Section 4), a lower value of $\alpha$ is appropriate. Note that $\alpha = 0$ will always try to pin the norms of backpropped gradients from each task to be equal at $W$.

Equation 1 sets a desired target for our gradient norms, and we want to update our loss weights $w_i(t)$ to move gradient norms towards this target. To accomplish this, GradNorm is implemented as a loss function $L_{\text{grad}}$ which is just the L1 distance between actual gradient norms and the targets in Equation 1:

$$L_{\text{grad}}^{(i)}(t; W) = |\text{norm}(\nabla_W L_i(t)) - E_{\text{task}}[\text{norm}(\nabla_W L(t))] \left( \frac{L_i'(t)}{E_{\text{task}}[L'(t)]} \right)^{\alpha} |. \tag{2}$$

The above loss is for one task; the full loss is just the mean of the individual task losses, $L_{\text{grad}}(t; W) = (1/T) \sum_i L_{\text{grad}}^{(i)}(t; W)$. $L_{\text{grad}}$ is then differentiated with respect to each $w_i(t)$, and its

---

[1]In general, if $L$ is a L2 or CE loss, one may instead prefer a loss $\phi(L)$ for some invertible function $\phi$. In that case, the inverse training rate should be set to $\phi^{-1}(L_i')$ to retain consistency. An L1 loss, for example, would use $(L_i')^2$ as a measure of inverse training rate.

gradients are applied via standard update rules to update these weights (see Figure 1 for a schematic view). In principle, it is also possible to update all network weights (not just $w_i(t)$) based on gradient of $L_{\mathrm{grad}}$, but in practice this adds undue complexity to the problem and often degrades performance.

We can choose $W$, the weights upon which we rate balance gradient norms, to be any subset of weights within layers of our network. In practice, in order to save on compute overhead, we choose $W$ to be the weights in the last layer which is shared amongst all three tasks. This simplification greatly shortens the number of layers $L_{\mathrm{grad}}$ must be backpropagated through, and with this choice of $W$ in our experiments GradNorm only adds $\sim 5\%$ of additional compute time. After every update step, we also renormalize the weights $w_i(t)$ so that $\sum_i w_i(t) = T$ in order to decouple gradient normalization from the global learning rate.

## 4   A SIMPLE TOY EXAMPLE

To illustrate GradNorm on a simple system, we consider $T$ regression tasks onto the functions

$$f_i(\mathbf{x}) = \sigma_i \tanh((B + \epsilon_i)\mathbf{x}), \tag{3}$$

where tanh acts element-wise. We use squared loss to train each task. The matrices $B$ and $\epsilon_i$ have elements generated IID from $\mathcal{N}(0, 10)$ and $\mathcal{N}(0, 3.5)$, respectively. Our task is thus to perform regression on multiple tasks with shared information $B$ along with information specific to each task, $\epsilon_i$. The $\sigma_i$ are fixed scalars which set the variance of the outputs $f_i$. Higher values of $\sigma_i$ induce higher values of squared loss for that task. These tasks are harder to learn due to the higher variances in their response values, but they also backpropagate larger gradients. Classically, such a scenario can lead to suboptimal training dynamics as the higher $\sigma_i$ tasks tend to dominate the training.

All toy problem runs use a 4-layer fully-connected ReLU-activated network with 100 neurons per layer as a common trunk. A final affine transformation per task gives $T$ final predictions. Inputs are in $\mathbb{R}^{250}$, and outputs lie in $\mathbb{R}^{100}$. To ensure consistency, we only compare models initialized to the same random values and fed data generated from a fixed random seed. The asymmetry $\alpha$ is set low to 0.12 for these experiments, as the output functions $f_i$ are all of the same form.

In these toy problems, we measure the *task-normalized* test-time loss, which is the sum of the test loss ratios for each task, $\sum_i L_i'(t)$. A simple sum of losses is wholly inadequate to judge the overall performance of a multitask network, as it biases itself towards tasks with higher loss scales, and there exists no general metric by which to judge multitask performance in any setting. Luckily, our toy problem was designed with tasks which are statistically identical except for their loss scales $\sigma_i$. For this simple example, there is therefore a clear measure of overall network performance, which is the sum of losses with each loss normalized to its $\sigma_i$ - precisely the sum of loss ratios.

In the case of $T = 2$, we choose the values $(\sigma_0, \sigma_1) = (1.0, 100.0)$. Classically, task 1 can suppress task 0's influence during training due to its higher loss scale. As shown in the top panels of Figure 2, gradient normalization remedies the issue by increasing $w_0(t)$ to counteract the larger gradients coming from $T_1$, and the improved task balance results in better test-time performance.

The possible benefits of gradient normalization become even clearer when the number of tasks increases. For $T = 10$, we sample the $\sigma_i$ from a normal distribution and plot the results in the bottom row of Figure 2. GradNorm significantly improves test time performance over naively weighting each task the same. Like $T = 2$, for $T = 10$ the $w_i(t)$ grow larger for smaller $\sigma_i$ tasks; GradNorm is giving tasks with smaller loss scales more breathing room.

For both $T = 2$ and $T = 10$, GradNorm is more stable and outperforms the uncertainty weighting proposed by Kendall et al. (2017). Uncertainty weighting, which enforces that $w_i(t) \sim 1/L_i(t)$, tends to grow weights too large and too quickly as the loss for each task drops. Although such networks train quickly at the onset, the training soon crashes as the global learning rate grows too large. This issue is exacerbated as uncertainty weighting allows $w_i(t)$ to change unconstrained (compared to GradNorm which ensures $\sum w_i(t) = T$ always), which pushes global learning rate up even further.

Overall, the traces for each $w_i(t)$ during a single GradNorm run seem fairly stable and convergent. In fact, in Section 5.3 we will see how the time-averaged weights $E_t[w_i(t)]$ lie close to the optimal static weights, suggesting GradNorm can greatly simplify the tedious grid search procedure.

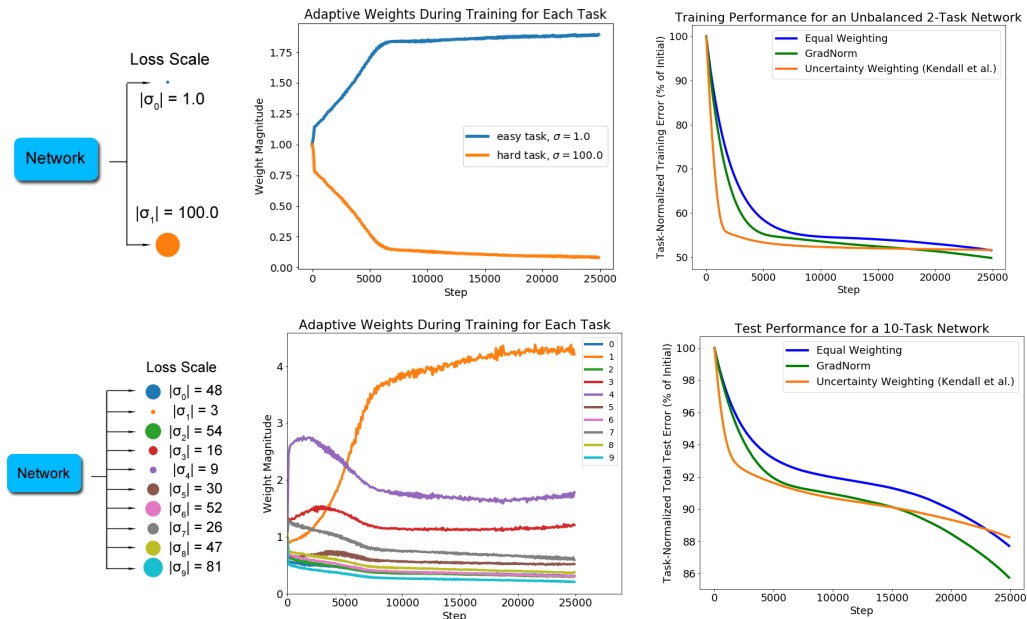

Figure 2: **Gradient Normalization on a toy 2-task (top) and 10-task (bottom) system.** Diagrams of the network structure with loss scales are on the left, traces of $w_i(t)$ during training in the middle, and task-normalized test loss curves on the right. $\alpha = 0.12$ for all runs.

## 5 APPLICATION TO A LARGE REAL-WORLD DATASET

We primarily use NYUv2 as our dataset of choice. The standard NYUv2 dataset carries depth, surface normals, and semantic segmentation labels (which we cluster into 13 distinct classes). NYUv2 is quite small as a dataset, with a training split of ∼800 examples, but contains both regression and classification labels, making it a good choice to test the robustness of GradNorm.

To show GradNorm in action on a more large-scale multitask dataset, we also expand NYUv2 to 40,000 images complete with pixel-wise depth, surface normals, and room keypoint labels. Keypoint labels are obtained through professional human labeling services, while surface normals are generated from camera parameters and the depth maps through standard methods.

Following Lee et al. (2017), the state-of-the-art in room layout prediction, all inputs are downsampled to 320 x 320 pixels and outputs to 80 x 80 pixels. These resolutions also speed up training without compromising complexity in the inputs or labels.

### 5.1 MODEL AND INDIVIDUAL TASK LOSSES

We try two different models: (1) a SegNet (Badrinarayanan et al. (2015); Lee et al. (2017)) network with a symmetric VGG16 (Simonyan & Zisserman (2014)) encoder/decoder, and (2) an FCN (Long et al. (2015)) network with a modified ResNet-50 (He et al. (2016)) encoder and shallow ResNet decoder. The VGG SegNet reuses maxpool indices to perform upsampling, while the ResNet FCN learns all upsampling filters. The ResNet architecture is further thinned (both in its filters and activations) to contrast with the heavier, more complex VGG SegNet: stride-2 layers are moved earlier and all 2048-filter layers are replaced by 1024-filter layers. Ultimately, the VGG SegNet has 29M parameters versus 15M for the thin ResNet. Although we will focus on the VGG SegNet in our more in-depth analysis, by designing and testing on two extremely different network topologies we will further demonstrate that GradNorm is very robust to the choice of base model.

We use standard pixel-wise loss functions for each task: cross entropy for segmentation, squared loss for depth, and cosine similarity for normals. As in Lee et al. (2017), for room layout we generate Gaussian heatmaps for each of 48 room keypoint types and predict these heatmaps with a pixel-wise squared loss. Note that all regression tasks are quadratic losses (our surface normal prediction uses

| Model Type and Weighting Method | Depth Error (m) | Segmentation 100-mIoU (%) | Normals Error (1-\|cos\|) |
|---|---|---|---|
| VGG SegNet, Depth Only | 1.038 | - | - |
| VGG SegNet, Segmentation Only | - | 70.0 | - |
| VGG SegNet, Normals Only | - | - | **0.169** |
| VGG SegNet, Equal Weights | 0.944 | 70.1 | 0.192 |
| VGG SegNet, GradNorm Converged Weights | 0.939 | **67.5** | 0.171 |
| VGG SegNet, GradNorm $\alpha = 1.5$ | **0.925** | 67.8 | 0.174 |

Table 1: **Test error, 320x320 NYUv2 for GradNorm and various baselines.**

a cosine loss which is quadratic to leading order), allowing us to use the loss ratio $L_i'(t)$ of each task as a direct proxy for each task's inverse training rate.

## 5.2 NETWORK PERFORMANCE

In Table 1 we display the performance of GradNorm on the NYUv2 dataset (with input/output resolutions as described in Section 5). Specific training schemes for all NYUv2 models are detailed in Appendix A. We see that GradNorm improves the performance of all three tasks with respect to the equal-weights baseline (where $w_i(t) = 1$ for all $t,i$), and that GradNorm either surpasses or matches (within statistical noise) the best performance of single networks for each task. The GradNorm Converged Weights network is derived by calculating the GradNorm time-averaged weights $E_t[w_i(t)]$ for each task (e.g. by averaging curves like those found in Appendix B), and retraining a network with weights fixed to those values. GradNorm thus can also be used to extract good values for static weights. We pursue this idea further in Section 5.3 and show that these weights lie very close to the optimal weights extracted from exhaustive grid search.

| Model Type and Weighting Method | Depth Error (m) | Keypoint Error (%) | Normals Error (1-\|cos\|) |
|---|---|---|---|
| Thin ResNet FCN, Depth Only | 0.725 | - | - |
| Thin ResNet FCN, Keypoint Only | - | 7.90 | - |
| Thin ResNet FCN, Normals Only | - | - | 0.155 |
| Thin ResNet FCN, Equal Weights | 0.697 | 7.80 | 0.172 |
| Thin ResNet FCN, Unc. Weighting (Kendall et al. (2017)) | 0.702 | 7.96 | 0.182 |
| Thin ResNet FCN, GradNorm Converged Weights | 0.695 | 7.63 | 0.156 |
| Thin ResNet FCN, GradNorm $\alpha = 1.5$ | **0.663** | **7.32** | **0.155** |
| VGG SegNet, Depth Only | 0.689 | - | - |
| VGG SegNet, Keypoint Only | - | 8.39 | - |
| VGG SegNet, Normals Only | - | - | 0.142 |
| VGG SegNet, Equal Weights | 0.658 | 8.39 | 0.155 |
| VGG SegNet, Unc. Weighting (Kendall et al. (2017)) | 0.649 | 8.00 | 0.158 |
| VGG SegNet, GradNorm Converged Weights | 0.638 | **7.69** | **0.137** |
| VGG SegNet, GradNorm $\alpha = 1.5$ | **0.629** | 7.73 | 0.139 |

Table 2: **Test error, expanded 320x320 NYUv2 for GradNorm and various baselines.**

To show how GradNorm can perform in the presence of a much larger dataset, we also perform extensive experiments on the expanded NYUv2 dataset, which carries a factor of 50x more data. The results are shown in Table 2. As with the standard NYUv2 runs, GradNorm networks outperform other multitask methods, and either matches (within noise) or surpasses the performance of single-task networks.

Figure 3 shows test and training loss curves for GradNorm ($\alpha = 1.5$) and baselines on the expanded NYUv2 dataset for our VGG SegNet models. GradNorm improves test-time depth error by $\sim 5\%$, despite ending with much higher training loss. GradNorm achieves this by aggressively rate balancing the network (enforced by a high asymmetry $\alpha = 1.5$), and ultimately suppresses the depth weight $w_{\text{depth}}(t)$ to lower than 0.10 (see Appendix B for more details). The same trend exists for keypoint regression, and is a clear signal of network regularization. In contrast, the uncertainty weighting technique (Kendall et al. (2017)) causes both test and training error to move in lockstep, and thus is not a good regularizer. Only results for the VGG SegNet are shown here, but the Thin ResNet FCN produces consistent results.

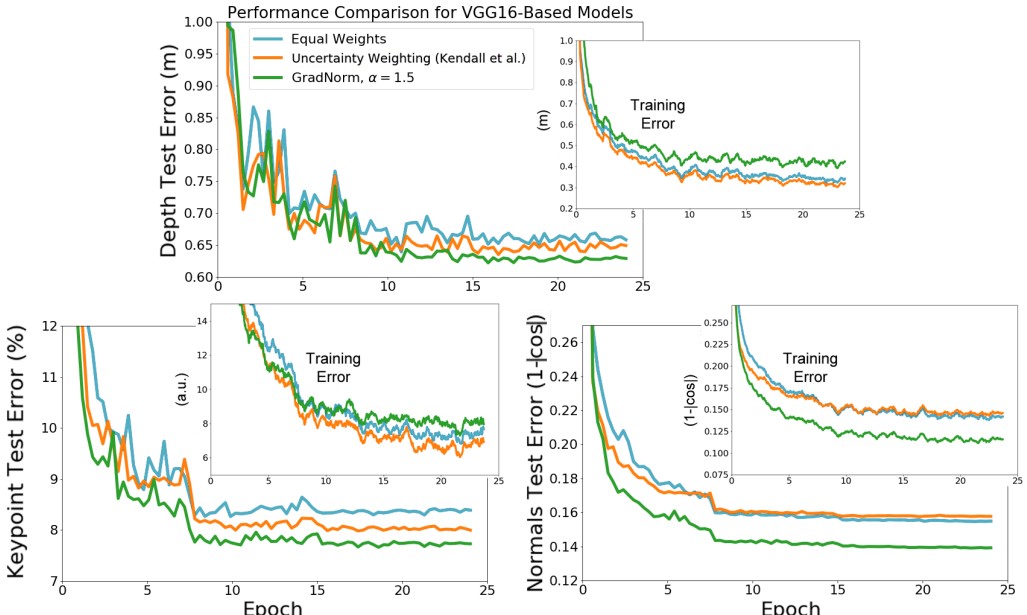

Figure 3: **Test and training loss curves for all tasks in expanded NYUv2, VGG16 backbone**. GradNorm versus an equal weights baseline and uncertainty weighting (Kendall et al. (2017)).

### 5.3 Gradient Normalization Finds Optimal Grid-Search Weights in One Pass

For our VGG SegNet, we train 100 networks from scratch with random task weights on expanded NYUv2. Weights are sampled from a uniform distribution and renormalized to sum to $T = 3$. For computational efficiency, we only train for 15000 iterations out of the normal 80000, and then compare the performance of that network to our GradNorm $\alpha = 1.5$ VGG SegNet network at the same 15000 steps. The results are shown in Figure 4.

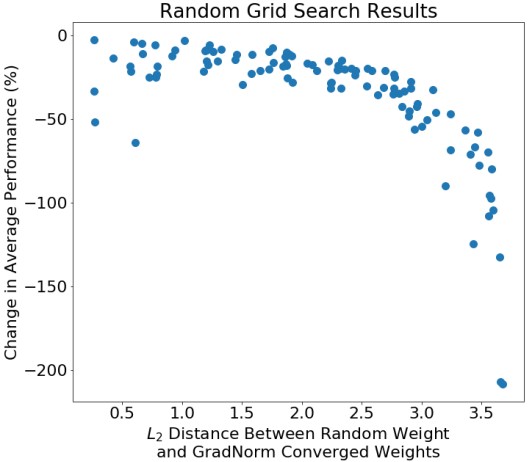

Figure 4: **Gridsearch performance for random task weights, expanded NYUv2**. Average change in performance across three tasks for a static network with weights $w_i^{\text{static}}$ is plotted against the $L_2$ distance between $w_i^{\text{static}}$ and our GradNorm network's time-averaged weights, $E_t[w_i(t)]$. All comparisons are made at 15000 steps of training.

Even after 100 networks trained, grid search still falls short of our GradNorm network. But even more remarkably, there is a strong, negative correlation between network performance and task weight distance to our time-averaged GradNorm weights. At an $L_2$ distance of $\sim 3$, grid search networks on average have almost double the errors per task compared to our GradNorm network. GradNorm has effectively allowed us to "cheat" and immediately find the optimal grid search weights without actually performing grid search, simplifying a process that is usually notoriously laborious.

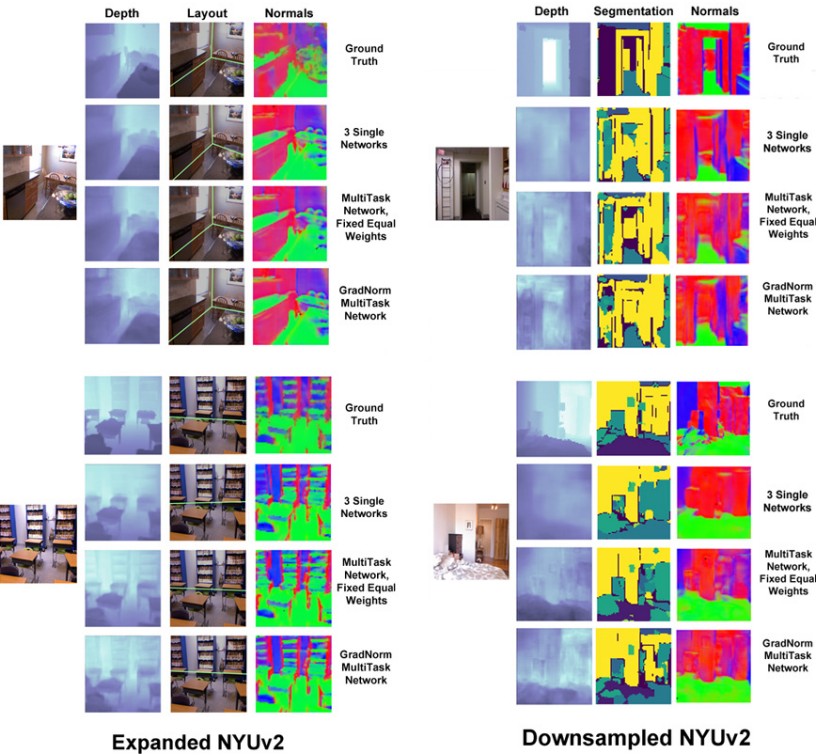

Figure 5: **Visualizations at inference time.** Expanded NYUv2 with room layout labels is shown on the left, while downsampled NYUv2 with semantic segmentation labels is shown on the right.

## 5.4 QUALITATIVE RESULTS

Figure 5 shows visualizations of the VGG SegNet outputs on test set images along with the ground truth, for both the expanded and downsampled NYUv2 datasets. Ground truth labels are juxtaposed with outputs from the equal weights network, 3 single networks, and our best GradNorm network. The qualitative improvements are incremental, but we find the GradNorm network tends to output smoother, more detailed pixel map predictions when compared to the other two baselines.

## 6 CONCLUSIONS

Gradient normalization acts as a good model regularizer and leads to superb performance in multitask networks by operating directly on the gradients in the network. GradNorm is driven by the attractively simple heuristic of rate balancing, and can accommodate problems of varying complexities within the same unified model using a single hyperparameter representing task asymmetry. A GradNorm network can also be used to quickly extract optimal fixed task weights, removing the need for exhaustive grid search methods that become exponentially more expensive with the number of tasks. We hope that our work has not only introduced a new methodology for quickly balancing multitask networks, but also has shown how direct gradient manipulation can be a powerful way to reason about task relationships within a multitask framework.

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

# Appendices

## A   GENERAL TRAINING CHARACTERISTICS

All runs are trained at a batch size of 24 across 4 Titan X GTX 12GB GPUs and run at 30fps on a single GPU at inference. NYUv2 runs begin with a learning rate of 2e-5. Expanded NYUv2 runs last 80000 steps with a learning rate decay of 0.2 every 25000 steps. Downsampled NYUv2 runs last 20000 steps with a learning rate decay of 0.2 every 6000 steps. Updating $w_i(t)$ is performed at a learning rate of 0.025 for both GradNorm and the uncertainty weighting (Kendall et al. (2017)) baseline. All optimizers are Adam, although we find that GradNorm is insensitive to the optimizer chosen. We implement GradNorm using TensorFlow v1.2.1.

## B   EFFECTS OF TUNING THE ASYMMETRY $\alpha$

The only hyperparameter in our technique is the asymmetry $\alpha$. The optimal value of $\alpha$ for NYUv2 lies near $\alpha = 1.5$, while in the highly symmetric toy example in Section 4 we used $\alpha = 0.12$. This observation reinforces why we call $\alpha$ an asymmetry parameter.

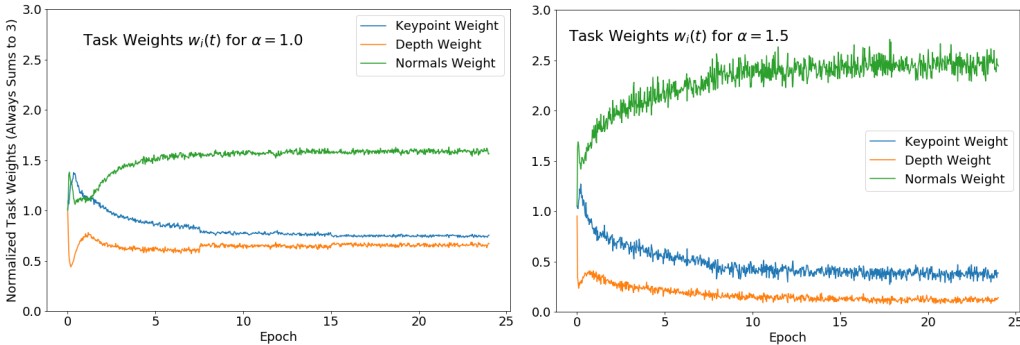

Figure 6: **Weights $w_i(t)$ during training, expanded NYUv2.** Traces of how the task weights $w_i(t)$ change during training for two different values of $\alpha$. A larger value of $\alpha$ pushes weights farther apart, leading to less symmetry between tasks.

Tuning $\alpha$ leads to performance gains, but we found that for NYUv2, almost any value of $0 < \alpha < 3$ will improve network performance over an equal weights baseline. Figure 6 shows that higher values of $\alpha$ tend to push the weights $w_i(t)$ further apart, which more aggressively reduces the influence of tasks which overfit or learn too quickly (in our case, depth). Remarkably, at $\alpha = 1.75$ (not shown) $w_{\text{depth}}(t)$ is suppressed to below 0.02 at no detriment to network performance on the depth task.

