# OpenReview forum: "GradNorm: Gradient Normalization for Adaptive Loss Balancing in Deep Multitask Networks"
_ICLR.cc/2018/Conference — Reject_

### Official Review · AnonReviewer1 · 2017-11-26
**Normalizing gradients for efficient multi-task network learning**

**Rating:** 6
**Confidence:** 2

**Review:**

The paper proposes a method to train deep multi-task networks using gradient normalization. The key idea is to enforce the gradients from multi tasks balanced so that no tasks are ignored in the training. The authors also demonstrated that the technique can improve test errors over single task learning and uncertainty weighting on a large real-world dataset.

It is an interesting paper with a novel approach to multi-task learning. To improve the paper, it would be helpful to evaluate the method under various settings. My detailed comments are below.

1. Multi-task learning can have various settings. For example, we may have multiple groups of tasks, where tasks are correlated within groups but tasks in different groups are not much correlated. Also, tasks may have hierarchical correlation structures. These patterns often appear in biological datasets. I am wondering how a variety of multi-task settings can be handled by the proposed approach. It would be helpful to discuss the conditions where we can benefit from the proposed method.

2. One intuitive approach to task balancing would be to weight each task objective based on the variance of each task.  It would be helpful to add a few simple and intuitive baselines in the experiments.

3. In Section 4, it would be great to have more in-depth simulations (e.g., multi-task learning in various settings). Also, in the bottom right panel in Figure 2, GrandNorm and equal weighting decrease test errors effectively even after 15000 steps but uncertainty weighting seems to reach a plateau. Discussions on this would be useful.

4. It would be useful to discuss the implementation of the method as well.

---

> ### Author Response · Authors · 2017-12-08
> **Re: Reviewer 1 Comments**
>
> General comments: Thank you very much for your review. You raise a very important point on task groupings and possible extensions to this method. Hopefully we can help clarify a few of these points below, along with the other comments/clarification requests you made.
>
> (((It is an interesting paper with a novel approach to multi-task learning. To improve the paper, it would be helpful to evaluate the method under various settings. My detailed comments are below.
>
> 1. Multi-task learning can have various settings. For example, we may have multiple groups of tasks, where tasks are correlated within groups but tasks in different groups are not much correlated. Also, tasks may have hierarchical correlation structures. These patterns often appear in biological datasets. I am wondering how a variety of multi-task settings can be handled by the proposed approach. It would be helpful to discuss the conditions where we can benefit from the proposed method.)))
>
> RESPONSE: We completely agree that this is a very important question. However, this type of task grouping is best handled through network architecture search, not through tuning the loss function. Network branch structure should begin to mimic correlations amongst network tasks (see for example Lu et al 2016, Farley et al 2015). There are certainly some exciting possibilities in the direction of using gradients for architecture search, but they’re rather out of the scope to the methods proposed in our manuscript, but are directions we are considering for future study.
>
> However, although GradNorm isn’t the most direct solution to correlation structures in the labels, we still do well in the presence of these correlations. In our NYUv2 experiments, there is actually a strong grouping amongst tasks: depth and normals are strongly correlated (in fact the latter is calculated from the former), while segmentation or room layout are less related and dependent on rather different semantics. Despite this, GradNorm still converges to optimal weights and improves performance on all tasks, due to the asymmetry $\alpha$: higher $\alpha$ informs our network to expect more complicated relationships between tasks, including complicated correlation structures.
>
> (((2. One intuitive approach to task balancing would be to weight each task objective based on the variance of each task.  It would be helpful to add a few simple and intuitive baselines in the experiments. )))
>
> RESPONSE: Kendall’s et al.’s methodology actually does precisely this. The Kendall et al. method (uncertainty weighting) is essentially a sophisticated variant of variance weighting - it uses a Bayesian framework to model intrinsic task variance, and then picks the loss weights w_i(t) based directly on these variance estimates. It is clear, however, that GradNorm outperforms this type of variance weighting from our experiments.
>
> (((3. In Section 4, it would be great to have more in-depth simulations (e.g., multi-task learning in various settings). Also, in the bottom right panel in Figure 2, GrandNorm and equal weighting decrease test errors effectively even after 15000 steps but uncertainty weighting seems to reach a plateau. Discussions on this would be useful.)))
>
> RESPONSE: In terms of more in-depth simulations, we did present results for a variety of different tasks: regression, classification, and synthetic/simulated. There are certainly more tasks we could try (although often we are limited by not having good multitask labels for various scenarios). We believe that experiments we performed show that our methodology is robust to many standard factors (architecture, single-task loss function choices, etc.)
>
> The pitfall of the uncertainty weighting technique comes from its tendency to overly boost gradients through training. As L_i decreases, uncertainty weighting aggressively tries to compensate with higher task weights, and a higher global learning rate as a result.  In our case, this improved training initially but then training reached a plateau when the global learning rate grew too large. We will make this clear in the revision.
>
> (((4. It would be useful to discuss the implementation of the method as well. )))
>
> RESPONSE: We welcome any specific detail requests, and we are planning to add many more details of the implementation in the revision (soon to come). We have also summarized the implementation of Gradnorm in RE: IMPLEMENTATION OF GRADNORM above in “Rebuttal: General Comments.”

---

> ### Author Response · Authors · 2017-12-19
> **Added paper revision**
>
> Hello,
>
> Thanks again for your feedback. We'd like to inform you that we have uploaded a paper revision which we feel provides a much clearer exposition of our technique. Time permitting, we invite you to take a look, in the hopes that this newer version clarifies any outstanding questions you may have had.

---

### Official Review · AnonReviewer3 · 2017-11-27
**Technique seems interesting and useful. But, the exposition of the technique is poor and several important details are missing.**

**Rating:** 4
**Confidence:** 4

**Review:**

Paper summary:
Existing works on multi-task neural networks typically use hand-tuned weights for weighing losses across different tasks. This work proposes a dynamic weight update scheme that updates weights for different task losses during training time by making use of the loss ratios of different tasks. Experiments on two different network indicate that the proposed scheme is better than using hand-tuned weights for multi-task neural networks.


Paper Strengths:
- The proposed technique seems simple yet effective for multi-task learning.
- Experiments on two different network architectures showcasing the generality of the proposed method.


Major Weaknesses:
- The main weakness of this work is the unclear exposition of the proposed technique. Entire technique is explained in a short section-3.1 with many important details missing. There is no clear basis for the main equations 1 and 2. How does equation-2 follow from equation-1? Where is the expectation coming from? What exactly does ‘F’ refer to? There is dependency of ‘F’ on only one of sides in equations 1 and 2? More importantly, how does the gradient normalization relate to loss weight update? It is very difficult to decipher these details from the short descriptions given in the paper.
- Also, several details are missing in toy experiments. What is the task here? What are input and output distributions and what is the relation between input and output? Are they just random noises? If so, is the network learning to overfit to the data as there is no relationship between input and output?


Minor Weaknesses:
- There are no training time comparisons between the proposed technique and the standard fixed loss learning.
- Authors claim that they operate directly on the gradients inside the network. But, as far as I understood, the authors only update loss weights in this paper. Did authors also experiment with gradient normalization in the intermediate CNN layers?
- No comparison with state-of-the-art techniques on the experimented tasks and datasets.


Clarifications:
- See the above mentioned issues with the exposition of the technique.
- In the experiments, why are the input images downsampled to 320x320?
- What does it mean by ‘unofficial dataset’ (page-4). Any references here?
- Why is 'task normalized' test-time loss as good measure for comparison between models in the toy example (Section 4)? The loss ratios depend on initial loss, which is not important for the final performance of the system.


Suggestions:
- I strongly suggest the authors to clearly explain the proposed technique to get this into a publishable state.
- The term ’GradNorm’ seem to be not defined anywhere in the paper.


Review Summary:
Despite promising results, the proposed technique is quite unclear from the paper. With its poor exposition of the technique, it is difficult to recommend this paper for publication.

---

> ### Author Response · Authors · 2017-12-08
> **Re: Reviewer 3 Comments**
>
> General comments: Thank you very much for your review. We are working on a revision that will address clarifications you asked for, but please allow us to respond to each of your points below.
>
> (((- The main weakness of this work is the unclear exposition of the proposed technique. Entire technique is explained in a short section-3.1 with many important details missing. There is no clear basis for the main equations 1 and 2. How does equation-2 follow from equation-1? Where is the expectation coming from? What exactly does ‘F’ refer to? There is dependency of ‘F’ on only one of sides in equations 1 and 2? More importantly, how does the gradient normalization relate to loss weight update? It is very difficult to decipher these details from the short descriptions given in the paper.)))
>
> RESPONSE: Please see “RE: IMPLEMENTATION OF GRADNORM” above in “Rebuttal: General Comments”.
>
> (((Also, several details are missing in toy experiments. What is the task here? What are input and output distributions and what is the relation between input and output? Are they just random noises? If so, is the network learning to overfit to the data as there is no relationship between input and output? )))
>
> RESPONSE: Our toy example illustrates a simple but important scenario where standard methods fail: multiple related regression tasks whose ground truth is statistically IID *except* for a scaling $\sigma_i$. The task was defined in equation (3) and described in the text directly following the equation. The target function is a multi-dimensional tanh function, and the inputs are $A_i = B + \epsilon_i$, with B a common baseline and individual elements of all matrices generated from a random normal distribution centered at 0 (B has std 10, while A_i std 3.5).
>
> (((Minor Weaknesses:
> - There are no training time comparisons between the proposed technique and the standard fixed loss learning.)))
>
> RESPONSE: This was mentioned in the manuscript, but GradNorm adds around 5% compute time to our networks. This is because we apply GradNorm only at a very upstream set of kernel weights.
>
> (((Authors claim that they operate directly on the gradients inside the network. But, as far as I understood, the authors only update loss weights in this paper. Did authors also experiment with gradient normalization in the intermediate CNN layers?)))
>
> RESPONSE: By “operate directly” on the gradients, we mean that the gradients are explicitly a part of our loss function (which necessitates taking gradients of gradients). This is in contrast with the traditional methods that do not explicitly take the first-order gradients into account. We chose to apply GradNorm only to a very upstream CNN layer because it saved on overhead compute significantly.
>
> (((No comparison with state-of-the-art techniques on the experimented tasks and datasets.)))
>
> RESPONSE: We did make a crucial comparison to the state-of-the-art: we showed that our multi-task balancing method produces superior results to the Kendall et al dynamic weighting method, which is a state-of-the-art method for multitask learning.
>
> After the submission we performed some tests on the full-resolution NYUv2 task on the standard dataset: GradNorm in that case improves depth error by ~10%, segmentation mIoU by ~7%, and normals error by ~28% over an equal weights baseline, so the results are consistent across resolutions.
>
>
> (((Clarifications:
> - See the above mentioned issues with the exposition of the technique.
> - In the experiments, why are the input images downsampled to 320x320?)))
>
> RESPONSE: Please see “RE: DATASET/SETTING USED” above in “Rebuttal: General Comments.”
>
> (((What does it mean by ‘unofficial dataset’ (page-4). Any references here?)))
>
> This description is confusing and will be removed from the revision, but in addition to running GradNorm on the standard NYUv2 dataset, we used an expanded version of NYUv2 with additional annotations that were labeled/calibrated in house (hence ‘unofficial’). This involved 40x increase in number of labels compared to NYUv2, and we would be happy to make this dataset available at the conference.
>
> (((Why is 'task normalized' test-time loss as good measure for comparison between models in the toy example (Section 4)? The loss ratios depend on initial loss, which is not important for the final performance of the system.)))
>
> Please see “RE: TOY EXAMPLE AND THE SUM-OF-LOSS-RATIO METRIC” above in “Rebuttal: General Comments.”
>
> (((Suggestions:
>
> - The term ’GradNorm’ seem to be not defined anywhere in the paper.)))
>
> We will make this more explicit in the revision.

---

> ### Author Response · Authors · 2017-12-19
> **Added revision**
>
> Hello,
>
> Thanks again for your comments. We wanted to let you know that we have uploaded a paper revision with significant rewrites for clarity, and have rewritten Section 3 entirely. We hope that this newer version presents GradNorm in a much more clear way and motivates why it works so well in a multitask setting.

---

### Official Review · AnonReviewer2 · 2017-12-04
**Lacking in clarity; experiments not convincing**

**Rating:** 4
**Confidence:** 4

**Review:**

The paper addresses an important problem in multitask learning. But its current form has several serious issues.

Although I get the high-level goal of the paper, I find Sec. 3.1, which describes the technical approach, nearly incomprehensible. There are many things unclear. For example:

-  it starts with talking about multiple tasks, and then immediately talks about a "filter F", without defining what the kind of network is being addressed.

- Also it is not clear what L_grad is. It looks like a loss, but Equation 2 seems to define it to be the difference between the gradient norm of a task and the average over all tasks. It is not clear how it is used. In particular, it is not clear how it is used to "update the task weights"

- Equation 2 seems sloppy. “j” appears as a free index on the right side, but it doesn’t appear on the left side.

As a result, I am unable to understand how the method works exactly, and unable to judge its quality and originality.

The toy experiment is not convincing.

- the evaluation metric is the sum of the relative losses, that is, the sum of the original losses weighted by the inverse of the initial loss of each task. This is different from the sum of the original losses, which seems to be the one used to train the “equal weight” baseline. A more fair baseline is to directly use the evaluation metric as the training loss.
- the curves seem to have not converged.

The experiments on NYUv2 involves non-standard settings, without a good justification. So it is not clear if the proposed method can make a real difference on state of the art systems.

And the reason that the proposed method outperforms the equal weight baseline seems to be that the method prevents overfitting on some tasks (e.g. depth). However, the method works by normalizing the norms of the gradients, which does not necessarily prevent overfitting — it can in fact magnify gradients of certain tasks and cause over-training and over-fitting. So the performance gain is likely dataset dependent, and what happens on NYU depth can be a fluke and does not necessarily generalize to other datasets.

---

> ### Author Response · Authors · 2017-12-08
> **Re: Reviewer 2 Comments**
>
> General comments: Thank you very much for your comments. We will upload a revised version of the manuscript with a reworked section 3.1 and a more detailed exposition: we hope this will make the methodology and motivations clearer. We also hope to clarify a few things regarding your other points below:
>
> (((Although I get the high-level goal of the paper, I find Sec. 3.1, which describes the technical approach, nearly incomprehensible. There are many things unclear. For example:
>
> -  it starts with talking about multiple tasks, and then immediately talks about a "filter F", without defining what the kind of network is being addressed.
>
> - Also it is not clear what L_grad is. It looks like a loss, but Equation 2 seems to define it to be the difference between the gradient norm of a task and the average over all tasks. It is not clear how it is used. In particular, it is not clear how it is used to "update the task weights")))
>
> RESPONSE: Please see “GENERAL COMMENTS ON IMPLEMENTATION” above in “Rebuttal: General Comments.”
>
> (((- Equation 2 seems sloppy. “j” appears as a free index on the right side, but it doesn’t appear on the left side. )))
>
> RESPONSE: The revision will have cleaner notation.
>
> (((As a result, I am unable to understand how the method works exactly, and unable to judge its quality and originality.
>
> The toy experiment is not convincing.
>
> - the evaluation metric is the sum of the relative losses, that is, the sum of the original losses weighted by the inverse of the initial loss of each task. This is different from the sum of the original losses, which seems to be the one used to train the “equal weight” baseline. A more fair baseline is to directly use the evaluation metric as the training loss. )))
>
> RESPONSE: Please see “RE: TOY EXAMPLE AND THE SUM-OF-LOSS-RATIO METRIC” above in “Rebuttal: General Comments.”
>
> (((- the curves seem to have not converged.)))
>
>
> RESPONSE: For ease of visualization we only show the first 25k steps of training, but the trend is consistent beyond that point as well.
>
> (((The experiments on NYUv2 involves non-standard settings, without a good justification. So it is not clear if the proposed method can make a real difference on state of the art systems. )))
>
>
> RESPONSE: Regarding the non-standard settings, please see “RE: DATASET/SETTING USED” above in “Rebuttal: General Comments.”
>
> In addition, we’ve already shown that GradNorm is optimal in some important ways (being able to find optimal grid search weights, for example) for task weight tuning, so any system which stands to benefit from properly tuned task weights should benefit from GradNorm, regardless of how complex or how many parallel components are in the model otherwise. Note that many of the architectures we used (VGG-SegNet and ResNet-FCN) are popular state-of-the-art architectures and we showed significant improvement in both.
>
> (((And the reason that the proposed method outperforms the equal weight baseline seems to be that the method prevents overfitting on some tasks (e.g. depth). However, the method works by normalizing the norms of the gradients, which does not necessarily prevent overfitting — it can in fact magnify gradients of certain tasks and cause over-training and over-fitting. So the performance gain is likely dataset dependent, and what happens on NYU depth can be a fluke and does not necessarily generalize to other datasets. )))
>
> RESPONSE: GradNorm should not magnify gradients on overfitting tasks. Overfitting will to artificially high training rates, and GradNorm will curtail gradients for tasks with high training rates. For NYUv2 this was very apparent in depth regression.
>
> It is true that we focused on the NYUv2 dataset, but we feel there are a few very strong reasons to believe that the results are very statistically significant and not due to dataset bias: (1) We tried on different subsets of tasks: one with segmentation and one with room layout regression. GradNorm showed consistent performance on both. (2) We also tried on very different architectures with various connectivities, and different dataset sizes. (3) GradNorm quickly converged to optimal gridsearch task weights and beat 100 randomly initialized static networks. It is highly unlikely that GradNorm would have arrived at this set of weights through random chance.

---

> ### Author Response · Authors · 2017-12-19
> **Added revision**
>
> Hello,
>
> Thanks again for your review. We wanted to inform you that we have uploaded a paper revision with significant rewrites for clarity, especially in Section 3 (which has essentially been rewritten). We hope that this newer version presents a much clearer case for why GradNorm is an intuitive and powerful way to improve multitask learning.

---

### Public Comment · (anonymous) · 2017-11-27
**The name of method**

Hi,

I am a bit confused why the method is called gradient NORMALIZATION? From my understanding, it is essentially dynamic weighted average according to eqn (1)(2). Am I correct?

In fact, the name "gradient normalization" was proposed earlier in the following paper:
https://arxiv.org/pdf/1707.04822.pdf

It might be good to clarify this to avoid any confusion.

Best.

---

> ### Author Response · Authors · 2017-11-29
> **RE: The name of method**
>
> Hi there,
>
> Thanks very much for the comment. You're absolutely right - GradNorm at an implementation level amounts to dynamically finding the right weights w_i(t) which then goes into a weighted average of each individual task loss. This is implemented via the equations in the manuscript. However, the core reason these equations are meaningful is because they set a common scale for our backpropped gradients by *normalizing* gradients to two additional pieces of data: (1) the average gradient norms amongst different tasks, and (2) the relative training rate of tasks. The influence of the latter is controlled by the asymmetry parameter alpha, as described in the manuscript. That's why we are normalizing our gradients: we discovered a meaningful common scale for these gradients which tells us how they relate to each other and we use these relationships to our advantage during training.
>
> Thanks also for pointing out the other paper - I think it would be more well-defined to just refer to our method as GradNorm, which we also used to draw the analogy to BatchNorm. Normalizing gradients can mean many different things (depending on the objective, the scale/data you normalize to, etc.), and our proposed method focuses on one way to do this in the context of multitask learning. But depending on the application there can certainly be other methods where the term "gradient normalization" would also apply.

---

### Author Response · Authors · 2017-12-08
**Rebuttal: General Comments**

Thanks to all the reviewers for your comments. We are in process of revising the manuscript to address all concerns but also would like to offer clarifications here to the questions/remarks we received.

-----

RE: IMPLEMENTATION OF GRADNORM

We received a few comments to clarify the implementation of GradNorm. We are in process of overhauling this explanation in the manuscript but give a short summary here of the method.

First, by filter F we mean kernel weights W for some layer in the network, and we will switch to this notation (both here and in the revision) for clarity. To summarize GradNorm:

We identify the kernel weights W for a layer in any neural network architecture. Usually this layer is the last layer which couples to all tasks within the network. We will normalize the gradients of the loss at W.
We want the norms of the gradients on W to be rate balanced (i.e. no task trains very quickly relative to other tasks). Therefore, the derivative of the task i loss w.r.t W denoted $|\nabla_W L_i|$ is made proportional to $[(r_i)^{-1}]^{\alpha}$ for relative task training rate $r_i$ and hyperparameter $\alpha$. We argued that the loss ratio $L’_i$ gives us the inverse task training rate, so $(r_i)^{-1} = L’_i/E_{task}[L’]$. (This is eq 1).
Since GradNorm only reasons about relative quantities, we should keep the mean gradient norm unchanged. The constant of proportionality in point (2) above is thus most naturally the average gradient norm, $E_{task}[|nabla_W L_i|]$. Eq 1 thus defines a target value for each gradient norm $|nabla_W L_i|$, and our method pushes gradient norms towards this target via an L_1 loss between the value of $|nabla_W L_i|$ versus the desired value. (This loss is eq 2).
We backpropagate this loss like any normal loss function into the loss weights w_i(t) of the network. In principle we could also backpropagate this signal into all network parameters but this tends to degrade performance and speed.

------

RE: TOY EXAMPLE AND THE SUM-OF-LOSS-RATIO METRIC.

There was some concern that we used the sum of loss ratios, $L_i(t)/L_i(0)$, as our performance metric for the toy example. Please see future revision for more discussion, but from a multitask perspective, designing an appropriate statistic by which to judge overall performance is very difficult. The toy example, however, involves tasks which are statistically IID except for a scaling factor $\sigma_i$ per task: thus, the sum of loss ratios is the natural choice for gauging the performance of the network.

For more complex real tasks, the sum of loss ratios may not be very meaningful. In NYUv2, the loss ratio weights are clearly not optimal (this is clear from our gridsearch experiments). If we knew how to pick the correct multitask evaluation metric in general, onerous methods like gridsearch would be obsolete, as the evaluation metric would automatically set the correct training loss. So in our toy example, using the sum of loss ratios as the training loss would be rather circular; the entire point is to start with equal weighting (note that GradNorm also initializes task weights to be equal), and then to evaluate how well methods perform based on a “true” aggregate performance metric.

-----

RE: DATASET/SETTING USED.

We received some comments that our training setting (image resolutions, etc.) seemed nonstandard. To clarify, we generally followed architectures and resolutions from Lee et al (2017) for room layout estimation, as it is the state-of-the-art for room layout estimation. This resolution also allowed for faster training without losing complexity in the inputs or outputs. As our results are for multitask learning and our baseline comparisons are to other multitask learning techniques which are agnostic to the dataset settings chosen, we emphasize that our results are not dependent on the resolution of the data or the specific dataset.

---

### Decision · Program_Chairs · 2018-01-29
**ICLR 2018 Conference Acceptance Decision**

**Decision:**

Reject

**Comment:**

This paper proposes a way to automatically weight different tasks in a multi-task setting.  The problem is a bit niche, and the paper had a lot of problems with clarity, as well as the motivation for the experimental setup and evaluation.